# *Mycobacterium tuberculosis* Genotypes and Drug Susceptibility Test Results from Timor-Leste: A Pilot Study

**DOI:** 10.3390/genes13101733

**Published:** 2022-09-27

**Authors:** Nevio Sarmento, Ella M. Meumann, Helder M. Pereira, Constantino Lopes, Maria Globan, Charlotte Hall, Matthew Di Palma, Nicole Hersch, Kristy Horan, Anna P. Ralph, Joshua R. Francis

**Affiliations:** 1Menzies School of Health Research, Charles Darwin University, Darwin, NT 0810, Australia; 2National TB Reference Laboratory, National Health Laboratory, Dili, Timor-Leste; 3National Tuberculosis Control Program, Ministry of Health, Dili, Timor-Leste; 4Victorian Infectious Diseases Reference Laboratory, Peter Doherty Institute for Infection and Immunity, Melbourne, VIC 3000, Australia; 5Maluk Timor, Dili, Timor-Leste; 6Bairo Pite Clinic, Dili, Timor-Leste; 7The Peter Doherty Institute for Infection and Immunity, Melbourne, VIC 3000, Australia

**Keywords:** Tuberculosis (TB), culture, sensitivity, sequencing, east Timor

## Abstract

Tuberculosis (TB) is prevalent and a major public health problem in Timor-Leste. The government of Timor-Leste is prioritising the surveillance of TB and drug-susceptibility testing (DST) to understand the burden of TB and TB drug resistance in the country. Moreover, little is known about the origin of *Mycobacterium tuberculosis* (MTB) in Timor-Leste. This study reports MTB DST and sequencing for Timor-Leste. A pilot study was carried out in which a convenience sample of TB isolates from mucopurulent sputum collected from presumptive TB patients in the capital Dili between July and December 2016 was tested for phenotypic and genotypic evidence of drug resistance. Standard MTB culture was performed at the Timor-Leste National Health Laboratory (NHL). The MTB isolates were sent to the Victorian Infectious Diseases Reference Laboratory (VIDRL) in Australia for DST and sequencing. Overall, 36 MTB isolates were detected at the NHL; 20 isolates were recovered during sub-culturing at VIDRL. All 20 isolates were susceptible to rifampicin, isoniazid, pyrazinamide, and ethambutol, with no genotypic markers of resistance identified. On sequencing, lineage 4 was the most common. The results of this study provide a small snapshot of MTB diversity and resistance in an under-sampled region with very high TB incidence. Future investment in whole-genome sequencing capacity in Timor-Leste will make it possible to undertake further, more representative analyses that may be used to evaluate transmission dynamics and epidemiology of genotypic markers of resistance.

## 1. Introduction

Tuberculosis (TB) is a major public health problem and research priority in Timor-Leste [1]. Ten million people contracted TB globally in 2019, while around 1.2 million deaths were reported in HIV-negative persons [2]. The World Health Organization (WHO) estimated an annual incidence rate of 508/100,000 and mortality rate of 106/100,000 [3] in Timor-Leste. Many barriers to TB detection exist, including limited access to diagnostic tests. In many health facilities in Timor-Leste, the primary method for TB diagnosis was smear microscopy through its designated microscopy centers (DMCs) [4]. In 2012, the GeneXpert MTB/RIF (Cepheid, Sunnyvale, CA, USA) polymerase chain reaction (PCR) assay was introduced into the country. Since then, access to PCR testing has expanded, with the availability of 11 GeneXpert machines [5] in 8/13 municipalities, coordinated by the National Tuberculosis Control Program (NTP) and supported by the National TB Reference Laboratory (NTRL) at the National Health Laboratory (NHL).

The GeneXpert MTB/RIF is useful for timely diagnosis with improved accuracy over microscopy; however, culture and drug susceptibility testing (DST) remain the gold standard [6,7]. TB culture is routinely performed at NTRL, NHL, Timor-Leste following a positive test (either slide microscopy or positive PCR). The extent of TB drug resistance in Timor-Leste was poorly described prior to 2019. In 2019, a National Anti-Tuberculosis Drug Resistance Survey (DRS) was conducted, showing low rates of resistance to first-line drugs with multidrug-resistant TB (MDR-TB) making up less than 2% of culture-confirmed cases [5].

Genomic sequencing of *Mycobacterium tuberculosis* (MTB) is increasingly being used for antimicrobial resistance prediction, and in 2021, the World Health Organization published its first catalogue [8] of mutations associated with resistance in MTB. Genomic sequencing can also be used to investigate transmission clusters, to differentiate relapse from reinfection, to understand within-host diversity, and to understand global MTB phylogeography [9].

This pilot study aimed to test a convenience sample of TB isolates from Dili, Timor-Leste with phenotypic DST and genotypic DST, to inform future service planning for the NTP in Timor-Leste.

## 2. Materials and Methods

### 2.1. Mycobacterial Culture, Drug Susceptibility Testing and Genomic Sequencing

This was a 6-month pilot study involving tuberculosis culture, DST, and genotyping for Timor-Leste. The study included sputum samples collected between July and December 2016 from clinics and the national hospital in Dili, the capital of Timor-Leste. Mycobacterial culture was performed using standard laboratory techniques at NTRL. Each sputum sample was treated with NaOH and inoculated into 3% Ogawa Slant Medium [10]. The medium was then incubated at 37 °C for 8 weeks. Growth on the medium was observed each week. A convenience sample of MTB isolates from positive cultures during the 6-month period were stored for further analysis.

The MTB colonies were stored in Tryptic Soy Broth (TSB) + 20% glycerol at −80 °C for more than three years at NTRL, awaiting approval for shipment to Australia. In 2019, the isolates were shipped to the Mycobacterium Reference Laboratory at the Victorian Infectious Diseases Reference Laboratory (VIDRL) in Victoria, Australia, where re-culturing was attempted. DST was performed on any re-cultured isolates using the BACTEC Mycobacterial Growth Indicator Tube 960 system (Becton Dickinson) standard method [11]. DNA extraction from cultured isolates was performed using methods as described by Votintseva et al. [12]. Whole-genome sequencing of extracted DNA was performed at the Microbiological Diagnostic Unit Public Health Laboratory in Melbourne. Unique dual-indexed libraries were prepared using the Nextera XT DNA sample preparation kit (Illumina, San Diego, CA, USA) and libraries were sequenced on the Illumina NextSeq 500/550 with 150-cycle paired-end chemistry as outlined in the manufacturer’s protocols (https://www.illumina.com, accessed 1 November 2019).

### 2.2. Bioinformatic Analyses

Sequence Read Archive accessions, metadata and quality metrics for the Timor-Leste MTB genomes sequenced as part of this study are listed in Appendix A. An additional 43 genomes from TB cases born in Timor-Leste sequenced as part of a previous northern Australian study were included for context (Appendix A) [13,14]. Variants were called using Snippy v4.3.6 (https://github.com/tseemann/snippy, accessed 20 August 2022) with the H37Rv genome (GenBank accession NC_000962.3) as reference, with a minimum coverage of 10 reads and minimum fraction of variant bases of 90%. Repetitive regions were masked from the alignment [15]. TBProfiler v2.8.4 was used to assign sublineages and to identify mutations associated with antimicrobial resistance [16]. IQ-TREE v2.0.3 was used to infer a maximum-likelihood phylogenetic tree with a generalised time reversible model with 4 γ categories, 1000 ultrafast bootstrap approximation replicates and 1000 bootstrap approximate-likelihood ratio test replicates [17]. A pairwise single-nucleotide polymorphism (SNP) distance threshold of ≤12 SNPs was used to identify possible transmission [9].

### 2.3. Ethical Approval

Two ethics approvals were obtained for this study: (i) Human Research Ethics Committee of the Northern Territory Department of Health and Menzies School of Health Research (HREC-2014-2309) and (ii) Timor-Leste National Institute of Health “Instituto Nacional de Saúde—INS” (DEPS-2015-119).

## 3. Results

During the six-month period of the study, there were 36 sputum samples that were culture positive for MTB in the NTRL, each obtained from a unique adult patient with suspected pulmonary TB based on clinical signs and symptoms. Additional clinical data were not collected.

A total of 20 of the 36 MTB isolates submitted were re-cultured and available for further analysis. All 20 were susceptible to rifampicin, isoniazid, pyrazinamide, and ethambutol (Table 1). Nineteen of the twenty isolates were successfully sequenced. Isolate 21 was not sequenced. Consistent with the phenotypic susceptibility results, no mutations associated with antimicrobial resistance were identified in the study genomes. Two pairs of isolates (20 and 27; 22 and 24) were separated from each other by two SNPs and one SNP respectively, suggesting possible transmission.

A maximum likelihood phylogeny including the 19 study genomes and 43 context genomes is presented in Figure 1. Lineage 4 was the most common, accounting for 30/62 (48%) isolates, including 12 study isolates. The most common sublineage was 4.3.4.1 (15 genomes, including 8 study genomes). Lineage 1 (22/60 [35%], 4 study isolates) and lineage 2 (10/62 [16%], 3 study isolates) were the next most common. Among the 43 context genomes, one was genotypically multidrug-resistant, with a Ser450Tyr substitution in *rpoB* and a Ser315Thr substitution in *katG*. An additional five genomes were predicted to have isoniazid monoresistance, harbouring a Ser315Thr substitution in *katG* (3 isolates), a Ser315Asn substitution in *katG* (1 isolate), and an *inhA* promotor mutation C > T at position −15 (1 isolate).

## 4. Discussion

The recovered MTB study isolates were all phenotypically and genotypically susceptible to rifampicin, isoniazid, pyrazinamide, and ethambutol. These findings are consistent with data from the national anti-TB DRS, where low rates of resistance to first-line TB drug and MDR-TB were reported [5,18]. However, the small sample size and convenience sampling mean that resistance rates in Timor-Leste cannot be inferred from this study.

Only 36 sputum samples were culture-positive for MTB during this pilot study, which was conducted at a time when mycobacterial culture capacity was only just being established at the NTRL. Only 20/36 isolates were revived at VIDRL. TSB with 20% glycerol has been known to preserve MTB, which can usually be recovered well after storage at −80 °C [19], but the prolonged storage (>3 years) and unknown number of power outages experienced at NTRL may have impacted on the viability of the cultures.

This study is limited by the convenience sampling, the small sample size, and the age of the data, given that samples were collected in 2016. It is also limited by the lack of linked clinical data, which would have provided additional insights into the epidemiology of TB infections in Timor-Leste at the time. However, it does provide interesting insights into the diversity of MTB lineages in Timor-Leste, and further evidence of low rates of drug resistance. It also demonstrates the ability to store MTB isolates over long periods of up to three years and still successfully re-culture viable bacteria from some samples for further analysis, although the high rate of failure to re-culture suggests that shorter storage periods may be important for similar future work.

The ongoing high burden of TB in Timor-Leste and increasing global mobility mean that the introduction of and local development of resistance are likely. Prospective MTB genomic sequencing can aid with the identification of resistance mutations for surveillance and guide individual treatment, and can identify clusters and provide insight into transmission dynamics to guide the public health response. We conclude that further research into the epidemiology of TB in Timor-Leste is needed, as well as ongoing support for laboratory strengthening to ensure in-country availability of culture, phenotypic DST, and genomic testing. In combination, these can help to delineate chains of transmission, identify emerging patterns of drug resistance, and assist in obtaining better patient outcomes in this setting of high-burden TB.

## Figures and Tables

**Figure 1 genes-13-01733-f001:**
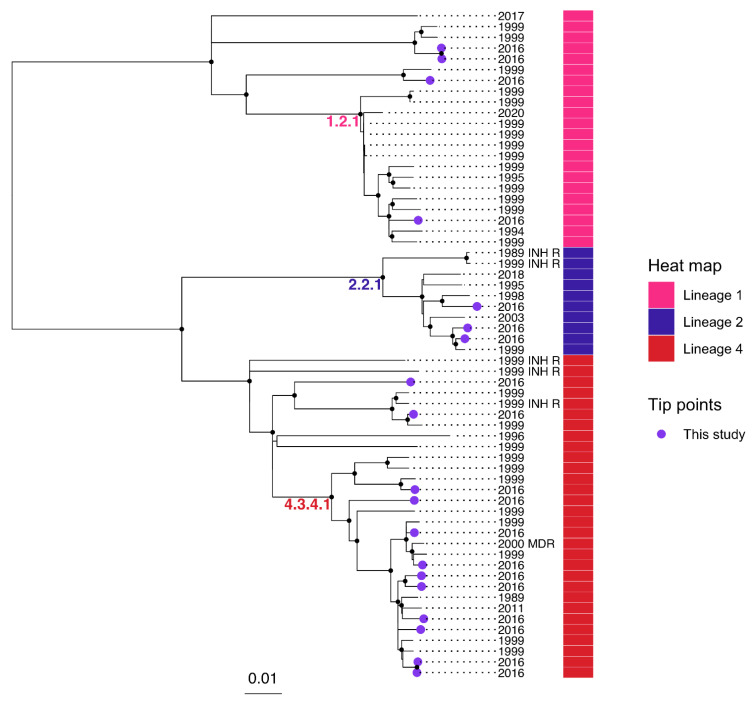
Midpoint-rooted maximum-likelihood phylogenetic tree including 19 study genomes from Timor-Leste, and 43 context genomes from TB cases born in Timor-Leste [14]. A sequence alignment including 8325 SNPs was used to infer the phylogeny. Nodes with approximate likelihood ratio >95 and ultrafast bootstrap >95 are marked with a black circle. Scale bar indicates substitutions/site. MDR, multidrug-resistant; INH R, isoniazid-monoresistant; based on mutations identified by TBProfiler [16].

**Table 1 genes-13-01733-t001:** Summary of culture and DST of 36 sputum isolates.

Isolate	Sublineage	Rifampicin	Isoniazid	Pyrazinamide	Ethambutol
1	NG	NG	NG	NG	NG
2	NG	NG	NG	NG	NG
3	4.3.3	S	S	S	S
4	4.4.1.2	S	S	S	S
5	NG	NG	NG	NG	NG
6	2.2.1	S	S	S	S
7	NG	NG	NG	NG	NG
8	1.2.1	S	S	S	S
9	NG	NG	NG	NG	NG
10	NG	NG	NG	NG	NG
11	NG	NG	NG	NG	NG
12	4.3	S	S	S	S
13	NG	NG	NG	NG	NG
14	4.3.4.1	S	S	S	S
15	NG	NG	NG	NG	NG
16	4.3.4.1	S	S	S	S
17	1.2.1	S	S	S	S
18	4.3.4.1	S	S	S	S
19	4.3.4.1	S	S	S	S
20	4.3.4.1	S	S	S	S
21	NS	S	S	S	S
22	1.2.2	S	S	S	S
23	NG	NG	NG	NG	NG
24	1.2.2	S	S	S	S
25	4.4.2	S	S	S	S
26	4.3.4.1	S	S	S	S
27	4.3.4.1	S	S	S	S
28	NG	NG	NG	NG	NG
29	2.2.1	S	S	S	S
30	4.3.4.1	S	S	S	S
31	2.2.1	S	S	S	S
32	NG	NG	NG	NG	NG
33	NG	NG	NG	NG	NG
34	NG	NG	NG	NG	NG
35	NG	NG	NG	NG	NG
36	NG	NG	NG	NG	NG

NG = no growth, means these isolates failed to regrow (re-culture) after arriving at VIDRL, Victoria, Australia; NS = not sequenced; S = susceptible to tested drugs; SRA = sequence read archive.

## Data Availability

The data presented in this study are available in the published manuscript and associated Appendix A.

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
