# Peer review of "Mycobacterium tuberculosis Genotypes and Drug Susceptibility Test Results from Timor-Leste: A Pilot Study"

_genes, 2022, doi:10.3390/genes13101733_

Round 1
Reviewer 1 Report
In this manuscript the authors made a description of the characteristics of twenty Tb isolates recovered from Dili, the capital of Timor Leste, and analyzed by WGS. The subject is important and could be relevant in the field; however, the work have several limitations that could limit its usefulness and need to resolved.
The incidence and mortality of TB in Timor Leste show that this is an important public health problem in the region. Unfortunately, no more epidemiological data is presented and by consequence is difficult the get a clearer idea about the situation of TB in the region, despite that, a recent DRS was made in the region, as the authors mentioned.
Methods
The sputum samples were from individuals with a suspicious of TB? How many samples were collected and confirmed cases of TB, and selected to be included in the study? Why clinical and epidemiological information of the patients was not recovered?
Why a phylogenetic analysis was not carried out only with the 19 sequenced isolates. Which were the conditions for the inclusion of the genomes that were used for the global phylogenetic analysis.
Why no analysis was performed to identify clonal or transmission complexes in the sequenced genomes.
Results
One paragraph with 6 lines is insufficient for the results section of one article.
Is necessary to include information about the composition of the individuals that conform the sample, correlation of phenotypic vs genotypic drug resistance, and the most important, the clinical data from the patients.
Close than 50% of the samples initially included in the study were lost in the processes, and no description or mention that explain this situation was mentioned, how this lost could influence the utility of WGS in the future.
The description of global phylogeny analysis was too broad and no mention of the association of the isolates sequenced with reference to the global data set was done. Why not search for specific associations with isolates from other regions, or for possible routes of transmission was done?
Discussion
P4L128-132, I agree with the authors in considering the small sample size and the lack of data as important limitations of the study, unfortunately I do not agree with the statement that "the data provide information about the diversity of lineages and the low rates of drug resistance".
The problems presented in relation to the shipment of samples is a variable that should be considered in depth, considering that this procedure wants to be incorporated into routine TB diagnostic practice in Timor Lester. In the same sense, it is necessary to carry out a cost study to determine the feasibility and benefit, considering that there is already culture and GeneXpert diagnostic studies in the area, and low incidence of drug resistance is reported.
Considering the small number of samples, the absent of drug resistant isolates and not conclusive phylogenetic studies, is difficult to agree with the authors when state that “these finding will provide support to the TB surveillance and promote the reduction of TB in the region”. Is necessary the development of additional studies considering the inclusion of more clinical isolates to have a better idea about the real utility of WGS in Timor-Lester.
Reviewer 2 Report
This interesting manuscript by Sarmento et al. uses WGS to classify and predict antimicrobial resistance within MTBC clinical isolates from Timor-Leste. Although the methodology is accurate and the main purpose of the manuscript is clear. The manuscript could be further improved by addressing the points as follows.
Major comments:
The main limitation of the study is the low number of samples analyzed (only 20/36), as well as the collection time of these samples (6 months), especially in a high incidence region such as Timor-Leste. Regarding this, the authors indicate that the low number of samples is due to the short collection time, however, taking into account the incidence of TB in the country (508/100,000 inhabitants), such sampling numbers are still very low. Only 36 cases were reported in 6 months. Could the authors give the total number of the Dili population, to obtain the incidence of the disease? Or could the authors explain why these numbers are very low.
This is a pilot study. I think this should be noted in the manuscript title as well as in the abstract section.
Interestingly, this high-incidence region has a low rate of resistant strains. Did the authors analyze the transmission?. Did they observe transmission related to neighbor countries?. How does immigration influence the incidence and as a consequence, the transmision of the disease?. This analysis will provide interesting insights into the local surveillance system and It will also give much more weight to the results presented here.
The authors included a global dataset in their analyses, however, there is poor information on this data set throughout the manuscript. Please add the total number of genomes used, as well as the studies from which they were obtained and their respective references. Are they genomes from neighboring or adjacent countries?. Please include the criteria followed to select these genomes.
The authors used the public platform “TB profiler” for the genomically resistance prediction, which is a good approach. However, do the authors use other public platforms such as “Mykrobe predictor” or “PhyResSe”?. It is known that not all platforms have the same catalog of mutations.
Please include a Supplementary table with the main WGS stats, for example, number reads, coverage obtained for each sample.
Minor comments:
Line 93: Please include the number of sites of the alignment used to infer the phylogeny.
Table 1: Why were you not able to sequence Isolate 21? Please add this information into the main text.
Figure 1: What does the blue and red background colors on the tree mean?.
Please add the number of genomes used to infer the phylogeny. In addition, please add the Drug resistance profile of each genome if available. Please add the Region Caption.
Round 2
Reviewer 1 Report
no comments
Author Response
Dear Reviewer 1,
Thank you for your feedback. We acknowledge that this is a pilot study and modifications are warranted. We submit the V2 below based on the suggestions and comments. Track changes and comments were made to highlight where the changes were made to address the comments.
Thank you very much,
Sincerely yours,
Nevio

Reviewer 2 Report
Comments
The authors have adequately discussed all the comments made by the reviewers. In addition, the authors have added relevant information in the main text highlighting the main results of their research.
I have a minor comment.
This is a pilot study. I think this should be noted in the manuscript title as well as in the abstract section.
Author Response
Dear Reviewer 2,
Thank you very much for your kind review. We have included the minor suggestions you requested. Attached is the V2 of the document including all the track changes and comments.
Thank you once again, and looking forward to hear your positive feedback.
Best regards,
Nevio
